# Identification and Verification of Biomarkers and Immune Infiltration in Obesity-Related Atrial Fibrillation

**DOI:** 10.3390/biology12010121

**Published:** 2023-01-12

**Authors:** Zhonghui Xie, Chuanbin Liu, Xu Lu, Zhijie Chen, Nan Zhang, Xinyan Wang, Xiaoqian Li, Yang Li

**Affiliations:** 1Medical School of Chinese PLA, Beijing 100853, China; 2Senior Department of Cardiology, The Sixth Medical Centre of Chinese PLA General Hospital, Beijing 100048, China; 3Western Medical Branch of the Chinese PLA General Hospital, Beijing 100853, China; 4Department of Cardiology, Fujian Provincial Hospital, Provincial Clinical Medicine College of Fujian Medical University, Fuzhou 350001, China

**Keywords:** atrial fibrillation, obesity, immune infiltration, inflammation, biomarkers

## Abstract

**Simple Summary:**

Obesity is an independent risk factor for atrial fibrillation, which, in the ensuing decades, will probably increase the global burden. Previous studies have indicated that inflammation is a central mediator between obesity and atrial fibrillation. However, the mechanisms underlying this crosstalk are still being uncovered, and there are insufficient specific biomarkers. We co-analyzed the atrial fibrillation and obesity microarrays to investigate the possible molecular mechanism of obesity-related atrial fibrillation. We found that *MNDA*, *CYBB*, *CD86*, *FCGR2C*, *NCF2*, *LCP2*, *TLR8*, *HLA-DRA*, *LCP1*, and *PTPN22* were only elevated in blood samples of obese atrial-fibrillation patients. Atrial-fibrillation patients’ left atrial appendage had increased infiltration of naïve B cells and decreased infiltration of memory B cells. Ten validated hub genes were related positively to naïve B cells and negatively to memory B cells. Ten validated genes identified by bioinformatics analysis, specifically correlated with obesity-related atrial fibrillation, may serve as biomarkers for obesity-related atrial fibrillation. These findings may also aid in comprehending pathophysiological mechanisms and identifying possible treatment targets for obesity-related atrial fibrillation.

**Abstract:**

Obesity is an independent risk factor for atrial fibrillation (AF). However, the mechanisms underlying this crosstalk are still being uncovered. Co-differentially expressed genes (co-DEGs) of AF and obesity microarrays were identified by bioinformatics analysis. Subsequently, functional enrichment, cell-type enrichment, and protein–protein interaction network analyses of co-DEGs were carried out. Then, we validated the hub genes by qRT-PCR of patients’ blood samples. Finally, CIBERSORT was utilized to evaluate the AF microarray to determine immune infiltration and the correlation between validated hub genes and immune cells. A total of 23 co-up-regulated DEGs in AF and obesity microarrays were identified, and these genes were enriched in inflammation- and immune-related function. The enriched cells were whole blood, CD33+ myeloid, and CD14+ monocytes. The hub genes were identified as *MNDA*, *CYBB*, *CD86*, *FCGR2C*, *NCF2*, *LCP2*, *TLR8*, *HLA-DRA*, *LCP1*, and *PTPN22*. All hub genes were only elevated in blood samples of obese-AF patients. The CIBERSORT analysis revealed that the AF patients’ left atrial appendage had increased infiltration of naïve B cells and decreased infiltration of memory B cells. The hub genes were related positively to naïve B cells and negatively to memory B cells. Ten hub genes may serve as biomarkers for obesity-related AF. These findings may also aid in comprehending pathophysiological mechanisms for obesity-related AF.

## 1. Introduction

Atrial fibrillation (AF) is one of the most common cardiac arrhythmias, while a study published in 2020 based on the prospective urban and rural epidemiological (PURE) study demonstrated that the incidence of AF is 270–360 cases per 100,000 [1]. Obesity was first identified as an independent risk factor for AF in the Framingham cohort study [2]. In recent years, an umbrella review and meta-analysis of observational and Mendelian randomization studies suggested that an increase of one unit in body mass index (BMI) was associated with an increased risk of AF [3]. Given that obesity has spread widely across the globe, it is likely to increase the global burden of AF in the coming decades. More seriously, obese patients have a lower success rate for catheter ablation and are more likely to relapse after surgery, while both BMI [4] and epicardial fat [5,6] are associated with the risk of AF progression and its recurrence. Therefore, it is necessary to investigate the potential pathogenesis of AF associated with obesity.

At present, the mechanisms involved in obesity-related AF remain unclear. Obesity may cause AF through local effects, such as abnormal secretion of adipokines, inflammation, hypoxia, and adipose and fibrous tissue infiltration [7,8], which may also increase the susceptibility to AF through systemic effects mechanisms such as insulin resistance, metabolism disorders, inflammatory states, and hemodynamic changes [9,10]. The mechanism appears diverse, but many studies proved that inflammation might be a pivotal mediator between obesity and AF [11]. However, there is still a lack of effective prediction methods for obesity-related AF, and the conditions and prognoses of these patients are not easily predictable. Therefore, it is necessary to search for specific biomarkers to identify the possible mechanism of obesity-related AF.

CIBERSORT is a widely used analysis tool, which can carry out deconvolution analysis of gene expression profiles and predict fractions of immune cell types in a mixed cell population [12]. Leukocyte signature matrix (LM22) can be used as the reference gene matrix for estimating 22 different types of immune cells. CIBERSORT is effective against numerous non-tumor diseases.

In this study, we used a microarray study as an effective method to explore possible pathogenesis and to find specific markers. Due to the lack of a microarray study of obesity-related AF patients, we analyzed the AF and obesity microarrays to investigate the co-DEGs and possible biomarkers of obesity-related AF. Then, we validated the hub genes in obesity-related AF patients’ blood samples. Finally, CIBERSORT was used to compare the immune infiltration of left atrial appendages (LAAs) between patients with sinus rhythm (SR) and AF. We investigated the relationship between hub genes and immune cell types further. Consequently, this study has important implications for identifying specific biomarkers and potential pathogenic mechanisms.

## 2. Materials and Methods

### 2.1. Materials and Methods

Figure 1 displays the study’s methodology. GSE79768 and GSE94752 were retrieved from Gene Expression Omnibus (GEO) database (http://www.ncbi.nlm.nih.gov/geo/, accessed on 26 April 2022). GSE79768 (platform: GPL570) has paired LAA and right atrial appendages (RAA) from 7 AF patients and 6 SR patients. GSE94752 (platform: GLP11532) has subcutaneous white adipose tissue (WAT) specimens from 9 lean and 39 obese patients. LAA data of the GSE79768 dataset were used to identify differentially expressed genes (DEGs) between patients with AF and SR. Data from GSE94752 was used to identify mRNA associated with obesity susceptibility. We first calculated the DEGs of these two microarrays and found that the up-regulated and down-regulated DEGs from these two microarrays intersected. By analyzing the co-DEGs of the two gene datasets listed above, we determined the potential relationship between AF and obesity. As described elsewhere, the AF patients in this study had a disease course lasting longer than one month, whereas the SR patients had no clinical evidence of AF [13]. We only analyzed the data of the left atrium because there is still evidence that the left atrium and right atrium have different gene expressions, and the left atrium plays a more critical role in the pathogenesis of AF [14]. The lean patients had a BMI of less than 25 kg/m^2^, while the BMI of obese patients was over 30 kg/m^2^, as indicated elsewhere [15].

### 2.2. Data Processing and Identification of DEGs and Co-DEGs

The raw datasets of GSE79768 and GSE94752 were accessed and quality controlled by R (Version 4.1.2) packages of “affy”, “affyPLM” and “limma”. Robust multiarray average (RMA) was used to normalize the data. The adjusted *p*-value < 0.05 and |log_2_FC| < 0.585 were used as thresholds to filter out DEGs. Hierarchical cluster heat maps were used to show distinguished mRNA expression levels of DEGs by the “pheatmap” package in R. Volcano plots were generated to represent all mRNA expression levels and the corresponding statistical inference value by “ggplot2” and “ggrepel” packages. Further, we calculated and made Venn diagrams for co-DEGs of AF- and obesity-DEGs by the “VennDiagram” package.

### 2.3. Gene Ontology, Pathways, and Cell-Type Enrichment Analyses of Co-DEGs

Gene ontology (GO) and Kyoto Encyclopedia of Genes and Genomes (KEGG) pathway enrichment analyses (http://www.kegg.jp/ or http://www.genome.jp/kegg/, accessed on 15 May 2022) of co-DEGs were carried out using the “clusterProfiler” and “pathview” packages of R software. Bar graphs were used to display enrichment results. The KEGG pathway graph was used to display the gene enrichment pathway. The q value (adjusted *p*-value) threshold was set to 0.05. Further, Enrichr (https://maayanlab.cloud/Enrichr/, accessed on 15 May 2022) and the Human Gene Atlas database were used to evaluate co-DEG cell types with enriched co-DEGs.

### 2.4. Protein–Protein Interaction Network Integration of Co-DEGs

Protein–protein interaction network integration (PPI) of co-DEGs was analyzed by the search tool for the retrieval of interacting genes (STRING database, Version 11.5; http://string-db.org/ accessed on 15 May 2022), which is suitable for identifying interactions between input genes. Subsequently, the analytic results of the STRING database were fed into Cytoscape software (Version 3.7.2). The biological networks and node degrees were analyzed and represented graphically, and the top ten hub genes were identified.

### 2.5. Validation for the Potential Role of Hub Genes

The expression levels of the top ten hub genes in obesity-related AF were confirmed by quantitative real-time polymerase chain reaction (qRT-PCR). This study included patients from the Chinese PLA General Hospital. To further investigate the function of hub genes, we obtained blood samples from eight lean-AF, eight obese-AF, seven lean-SR, and eight obese-SR patients. Patients’ clinical information can be seen in Appendix A. Total RNA from blood was prepared using a high-performance blood total RNA extraction kit (DP443, TIANGEN). Reverse RNA transcription was performed by a reverse-transcription system (K1622, Thermo Fisher, Waltham, MA, USA). Obtained cDNA was amplified using SYBR premix kit (A25742, Thermo Fisher) by BIO-RAD CFX96 (Bio-Rad Laboratories, Hercules, CA, USA). Predesigned gene-specific primers are listed in Appendix A. β-actin RNA levels were used as an internal control. This study protocol involving human subjects had been reviewed and approved by the Chinese People’s Liberation Army General Hospital Ethics Committee. The patients/participants provided their written informed consent to participate in the study. The AF patients were diagnosed with an electrocardiogram (ECG), whereas the SR patients did not exhibit any symptoms of AF (such as palpitations, dizziness, chest pain, or pressure) or ECG evidence. The lean patients’ BMI were no more than 25 kg/m^2^, and they did not have any metabolic disorders, for example, hyperlipidemia or diabetes mellitus. The patients who had a BMI greater than 30 kg/m^2^ were regarded as obese patients. 

### 2.6. Immune Infiltration by CIBERSORT Analysis of AF-DEGs

We used the CIBERSORT algorithm to analyze the normalized data generated from raw data, and the leukocyte signature matrix (LM22), which contains 22 types of immune cells, was used as the reference gene matrix [12]. LM22 includes macrophages M0, M1, and M2, memory B cells, B naïve cells, plasma cells, CD8 T cells, gamma-delta T cells, follicular helper T cells, CD4 memory-activated T cells, CD4 naïve T cells, CD4 memory resting T cells, regulatory T cells (Tregs), neutrophils, monocytes, activated NK cells, resting NK cells, resting dendritic cells, activated dendritic cells, eosinophils, activated mast cells, and resting mast cells. With a *p*-value less than 0.05, the samples were considered successful for deconvolution analysis and further subjected to differential analysis of proportions of immune cells. Further, “ggplot2”, “pheatmap,” and “vioplot” packages were utilized to visualize the percentage and difference of immune cell infiltration. The relationships between two immune cells and between immune cells and validated hub genes were calculated by the “corrplot” package.

### 2.7. Statistical Analysis

R (4.1.2) was utilized to conduct the bioinformatics analysis. This study’s qRT-PCR data were processed using GraphPad Prism (Version 9). Using F-tests, the expression levels of hub genes were analyzed. Adjusted *p*-value (q value) and *p*-value less than 0.05 were deemed statistically significant.

## 3. Results

### 3.1. Identification of DEGs in Atrial Fibrillation and Obesity

We identified 54,675 probes corresponding to 20,267 genes in the GSE79768 dataset and 33,298 probes corresponding to 19,409 genes in the GSE94752. There are 412 DEGs between AF patients and SR patients in GSE79768 LA specimens, including 292 up-regulated genes and 120 down-regulated genes. There are 746 DEGs between obese and lean patients in GSE94752 WAT specimens, including 547 up-regulated genes and 199 down-regulated genes. Following this, we discovered one co-down-regulated gene and twenty-three co-up-regulated genes in the aforementioned two datasets. The above data are available in Appendix A. Heat maps of the top 50 up-regulated and top 50 down-regulated genes of AF-DEGs and obese-DEGs are presented, and these figures can be found in Appendix A. Figure 2a illustrates the number of co-expressed genes of AF- and obese-DEGs, and these genes are labeled in Figure 2b, which displays the distribution of the differences in gene expressions. Further, Figure 2c,d demonstrate hierarchical clustering analysis of AF- and obese-related DEGs, respectively.

### 3.2. GO terms, KEGG Pathway, and Cell-Type Enrichment in Co-DEGs

Given that there is only one down-regulated gene in co-DEGs, only the up-regulated genes were utilized for enrichment analysis. The GO-terms enrichment results are shown in Figure 3a and Appendix A. The top ten GO terms related to biological processes (BPs) among these genes were: positive regulation of cytokine production, immune response-regulating signaling pathway, immune response-activating cell-surface-receptor signaling pathway, immune response-activating signal transduction, immune response-regulating cell-surface-receptor-signaling pathway, activation of the immune response, lymphocyte activation involved in immune response, positive regulation of α-β T cell activation, positive regulation of interferon-gamma (IFN-γ) production, and positive regulation of T cell activation. In terms of cellular components (CCs), there was a significant correlation in NADPH oxidase complex, MHC class II protein complex, MHC protein complex, an integral component of the lumenal side of endoplasmic reticulum membrane, the lumenal side of endoplasmic reticulum membrane, the endocytic vesicle, the lumenal side of the membrane, an integral component of endoplasmic reticulum membrane, an intrinsic component of endoplasmic reticulum membrane, and the lysosomal membrane. The terms related to molecular function (MF) mainly include superoxide-generating NAD(P)H oxidase activity, oxidoreductase activity, acting on NAD(P)H, oxygen as acceptor, MHC class II protein-complex binding, MHC protein-complex binding, peptide antigen binding, GTPase binding, oxidoreductase activity, acting on NAD(P)H, peptide-binding, electron-transfer activity, amide-binding.

The KEGG pathway enrichment results are shown in Figure 3b and Appendix A. In the KEGG analysis, the up-regulated genes were mainly enriched in leishmaniasis, phagosome, staphylococcus aureus infection, allograft rejection, graft-versus-host disease, type I diabetes mellitus, the intestinal immune network for IgA production, autoimmune thyroid disease, viral myocarditis, rheumatoid arthritis, hematopoietic cell lineage, asthma, osteoclast differentiation, systemic lupus erythematosus, cell adhesion molecules, tuberculosis, diabetic cardiomyopathy, inflammatory bowel disease, and antigen processing and presentation.

Enrichment analysis of cell types indicated that up-regulated genes in co-DEGs are more likely to identify whole blood, CD33+ myeloid, and CD14+ monocytes (Figure 3c and Appendix A) The interaction of enriched cell types is shown in Figure 3d, which shows an interaction among whole blood, CD33+ myeloid, and CD14+ monocytes. Further, the relationship between cell-type enrichment and genes is shown in Figure 3e. Figure 3f depicts the pathway map of the phagosome, which, according to the aforementioned enrichment analysis, may play a role in obesity-related AF.

### 3.3. PPI Network Analysis in Co-DEGs

As 21 co-DEGs filtered into the PPI network, we identified 16 nodes and 43 edges, and these data appear in Figure 4a. The hub genes calculated by Cytoscape software are *MNDA* (score 320), *CYBB* (score 314), CD86 (score 312), *FCGR2C* (score 242), *NCF2* (score 240), *LCP2* (score 182), *TLR8* (score 148), *HLA-DRA* (score 54), *LCP1* (score 30), and *PTPN22* (score 8), which are considered to be associated with obesity-related AF. (Figure 4b) Among these genes, *MNDA*, *CYBB*, and *CD86* possessed the top three scores, calculated by the maximal clique centrality (MCC) algorithm.

### 3.4. Validation for the Potential Role of Hub Genes

We confirmed hub-gene expression levels in lean-SR, obese-SR, lean-AF, and obese-AF patients’ blood samples. As depicted in Figure 5, all ten hub genes were uniquely expressed at a higher level in obese-AF patients compared with lean-SR, obese-SR, and lean-AF patients.

### 3.5. Immune Infiltration Analyses

Due to the co-DEGs being enriched in whole blood, CD33+ myeloid, and CD14+ monocytes, we further analyzed the DEGs in LAA samples between SR and AF patients by the CIBERSORT algorithm. The overall differential expressions of immune fractions between SR and AF patients are depicted in Figure 6a. More specifically, in Figure 6b, the LAA of AF patients exhibited a higher infiltration of naïve B cells and a lower infiltration of memory B cells compared to SR patients. As shown in Figure 6c, naïve B cells were negatively associated with memory B cells, regulatory T cells (Tregs), resting NK cells, M2 macrophages, and activated mast cells, whereas they were positively associated with gamma-delta T cells, M1 macrophages, and resting mast cells. Memory B cells were negatively associated with naïve B cells, CD4 memory resting T cells, gamma-delta T cells, M1 macrophages, resting mast cells, and neutrophils, while positively associated with Tregs, resting NK cells, M2 macrophages, activated dendritic cells, and activated mast cells. However, no correlation was found between monocytes, memory B cells, and naïve B cells. Therefore, we further analyzed the correlation between hub genes and immune cells. As illustrated in Figure 6d, naïve B cells were positively associated with *LCP1*, *HLA-DRA*, *TLR8*, *CD86*, and *CYBB*. Meanwhile, memory B cells were negatively associated with *PTPN22*, *LCP1*, *HLA-DRA*, *TLR8*, *LCP2*, *NCF2*, *FCGR2C*, *CD86*, *CYBB*, and *MNDA*. 

## 4. Discussion

In recent years, the incidences of atrial fibrillation (AF) and obesity have increased simultaneously, garnering more attention [16]. In obese patients, the adipose tissue (AT) is in a state of inflammation, which can cause AF [17], and recent studies have revealed that AF can be reduced by intervening in the “quality” of AT [18]. EAT is a highly active visceral tissue that produces many different pro-inflammatory adipokines that can diffuse directly to the myocardium and cause immune-cell infiltration and inflammation that may lead to AF [19]. The current study has identified and verified ten biomarkers for obesity-related AF by analyzing co-DEGs of AF microarray and obese adipose tissue microarray. In addition, we demonstrated a positive correlation between these biomarkers and the ratio of naïve B cells to memory B cells in the LAA of AF patients, which aids in the identification of initiating factors and potential therapeutic targets in obesity-related AF.

We found 23 co-DEGs in the two microarrays, 22 of which were up-regulated and 1 was down-regulated. First, based on a complete study of GO, KEGG, and cell-type enrichment of these co-up-regulated genes, we hypothesize that AT of obese patients and LAA of AF patients shares certain physiological processes. The blood’s CD33+ myeloid cells and CD14+ monocytes adhere to and infiltrate the tissues. The superoxide-generating activity of NADPH oxidase complex, phagocytosis activity, and MHCII antigen-presentation function of monocyte-derived macrophages was enhanced. In addition, the neutrophil extracellular trap formation was strengthened. Further, lymphocytes were activated in the tissues, and the secretion of cytokines(including IFN-γ) was enhanced. Second, the hub genes in co-up-regulated DEGs were identified as *MNDA*, *CYBB*, *CD86*, *FCGR2C*, *NCF2*, *LCP2*, *TLR8*, *HLA-DRA*, *LCP1*, and *PTPN22* by the PPI network. These hub genes were validated in blood samples of obesity-related AF. Third, the CIBERSORT analysis of the LAA of AF patients suggested increased infiltration of B cell naïve and decreased infiltration of memory B cells. The ratio of naïve/memory B cells had a positive association with CD4 memory resting T cells, gamma-delta T cells, M1 macrophages, resting mast cells, and neutrophils; and a negative association with Tregs, resting NK cells, M2 macrophages, activated dendritic cells, and activated mast cells. The validated hub genes were positively associated with the naïve/memory B cell ratio.

In obese patients, adipose tissue can release danger-associated molecular patterns (DAMPs) and then cause immune cell infiltration by Toll-like receptors (TLR) [20]. In 2008, Chen et al. first identified that AF patients have more inflammatory cells identified as CD45+ infiltrating the atria than normal controls [21]. Then, more research revealed that inflammation is an essential common pathway for AF caused by obesity [22]. This is consistent with our research. Co-DEGs were primarily enriched in immune function in our study. The ten hub genes pertain to the immune system. All ten hub genes were only raised in the blood of obese-AF patients, but not in the blood of lean-SR, lean-AF, or obese-AF patients. At present, few studies have found specific markers for obesity-related AF, while our research found that blood samples qRT-PCR of these genes may become a novel convenient method for predicting obesity-related AF.

*CYBB*, *FCGR2C*, *NCF2*, and *HLA-DRA* of validated hub genes were enriched in the phagosome pathway. *LCP1*, LCP2, *CD86,* and *HLA-DRA* were enriched in lymphocyte activation and immune response. *MNDA*, *CYBB*, *CD86*, *TLR8,* and *PTPN22* were enriched in positive regulation of cytokine production. Research published in 2023 in Science also suggests that a history of obesity reprograms mononuclear phagocytes, leading to transcription of pro-inflammatory cytokines [23]. Some studies have suggested that the inflammatory increase of macrophages and inflammatory factors can lead to AF through ion channel remodeling and structural remodeling [24,25]. The inflammatory-related genes are generally increased in classically activated macrophages (M1). Macrophages with increased NOX2 and enhanced phagocytosis can stimulate neutrophil extracellular trap formation [26]. In addition, this type of macrophage can not only secrete inflammatory cytokines, and enhance MHCII antigen presentation, but also stimulate the activation of CD4+ T cells and B cells [27], which is consistent with our study’s findings.

*CYBB* and *NCF2*, both NADPH oxidase (NOX), are present in phagocytic cells and participate in the “respiratory burst” to produce a large number of reactive oxygen species (ROS) and contribute to T cell activation [28]. The ROS can be released from the phagocytic cells and act on other cells [29]. ROS is one of the primary mechanisms causing obesity-related AF [22], and there exists a mechanism called “ROS-induced ROS release” (RIRR) [30]. Therefore, so finding the trigger source of ROS may play a critical role in obesity-related AF prevention. There has been some controversy about the role of NOX2 in the pathogenesis of AF. A 2015 study suggested intermittent hypoxia may induce AF by activating NOX2, thereby decreasing atrial Cx40 and Cx43 [31]. A study in 2020 suggested that NOX2 may increase the incidence of AF in obese mice by reducing I_Na_, I_Kur_, and atrial action potential duration (APD), and mitochondrial antioxidants could prevent the occurrence of AF [32]. However, another study in 2021 suggested that in atrial NOX2 overexpression mice, NOX2 overexpression could only slightly increase the induction rate of AF. However, no effect on electrophysiology and structural remodeling was found, and ROS induced by overexpression of human NOX2 in mouse myocardium is not the primary cause of the rising incidence of AF [33]. Atrial ROS main sources vary in AF courses and atrial matrix [34]. In our study, the increase of NOX2 in obesity was mainly concentrated in monocytes, so NOX2 and ROS causing AF may be mainly related to the increase of inflammatory-monocyte-derived macrophages rather than cardiomyocytes. Currently, studies on the co-effects of *NCF2* and *CYBB* on ROS are mainly seen in autoimmune, infectious, and ischemic diseases [35,36,37]. The mechanism of *NCF2* in AF needs to be further studied.

*HLA-DRA* and *CD86* are both biomarkers of M1 macrophages. A study based on human cardiomyocytes suggests that M1 macrophages with high expression of *CD86* in atrial fibrillation may promote extracellular matrix remodeling of atrial fibroblasts and participate in the occurrence of atrial fibrillation [38]. *TLR8* (toll-like receptor 8) recognizes pathogen-associated molecular patterns (PAMPs) and mediates the production of cytokines necessary for the development of effective immunity. A study reported that *TLR8* was correlated with levels of IL6, IL1β, and a greater inflammatory response, which is an important mechanism that causes atrial fibrillation [30]. However, the mechanism of *NCF2*, *HLA-DRA*, *TLR8*, *FCGR2C*, *MNDA*, *LCP1*, *LCP2*, and *PTPN22* in AF has not been studied. A comprehensive examination of these genes could aid in developing targeted protective measures to limit the damage caused by abnormal immunity.

In 2010, a study revealed the infiltration of CD3+ T cells and a small number of CD20+ B cells in the atria of patients with AF by the immunohistochemistry method [39], but the exact mechanism of lymphocyte involvement in AF has not been investigated fully. Our findings suggest a possibility that the activated lymphocytes can secrete lymphokines such as IFN-γ, further enhancing the phagocytosis and respiratory burst of monocytes [40]. It has been reported that in systemic sclerosis, the activation and apoptosis of the memory subset of B cells stimulate the proliferation of the naïve subset of B cells, resulting in an increased proportion of naïve/memory B cells [41], so a higher percentage of naïve/ memory B cells may be associated with a pro-inflammatory state. It has been suggested that obesity, pro-inflammatory factors, and phagocytosis-derived Fc ligand can cause the depletion of memory B cells [42,43], which may explain the increase in the ratio of naïve/memory B cells in this study. Furthermore, in this study, the pro-inflammatory immune cells like, CD4 memory resting T cells, gamma-delta T cells, M1 macrophages, resting mast cells, and neutrophils were positively correlated with the ratio of naïve/memory B cells. In addition, the expression of the validated hub genes is directly proportional to the naïve/memory B cells ratio and may be involved in regulating the infiltration and activation of B cells.

Still, our study has several limitations. First, we have tentatively verified the correlation among these hub genes, obesity, and AF in small-sample subjects. Therefore, the results of this study need to be further investigated in a cohort study with larger sample sizes. Second, this study only focused on one AF dataset due to limited datasets with intra-dataset distinction and correctable inter-dataset batch effect, so greater sample sizes of LAA samples from obese-AF patients should be further verified. Finally, more in vivo and in vitro studies containing protein level verifications are warranted to elucidate the underlying mechanisms among these validated genes, infiltrating immune cells and obesity-related AF.

## 5. Conclusions

In summary, we identified ten genes by bioinformatics analysis, which were specifically correlated with obesity-related AF, and these genes could be potential biomarkers for obesity-related AF. Our findings may potentially contribute to a deeper understanding of the pathophysiological mechanism and the identification of possible treatment targets in obesity-related AF.

## Figures and Tables

**Figure 1 biology-12-00121-f001:**
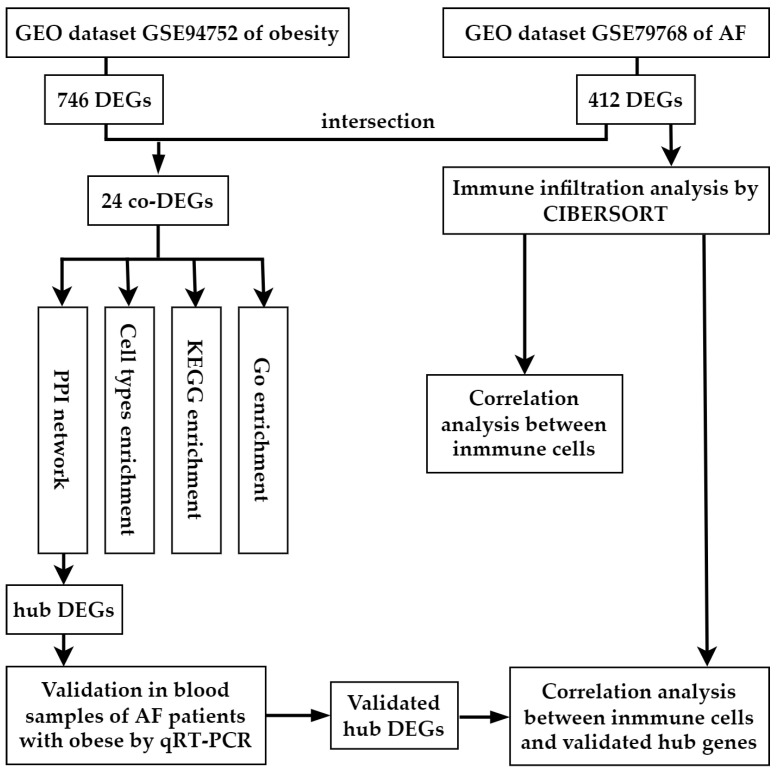
Workflow of the bioinformatics analysis methods in the present study. GEO, Gene Expression Omnibus; AF, arial fibrillation; DEGs, differentially expressed genes.

**Figure 2 biology-12-00121-f002:**
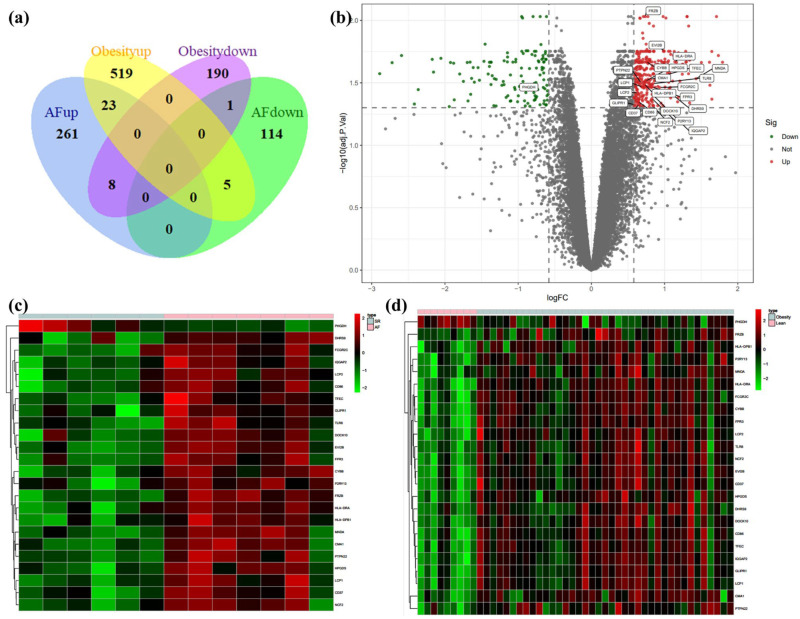
Venn diagram, volcano plot, and heat map of AF- and obese-related DEGs. (**a**) Venn diagram of DEGs related to AF and obesity. (**b**) Volcano plot of DEGs related to AF. Gray points represent the adjusted *p*-value > 0.05; red points represent adjusted *p*-value > 0.05 and up-regulated genes; green points represent adjusted *p*-value < 0.05 and down-regulated genes. (**c**) Heat map of GSE79768 with co-DEGs related to AF and obesity. The vertical axis represents samples, and the horizontal axis represents co-DEGs. Pink, AF samples; blue, SR samples; red, greater expression; green, less expression. (**d**) Heat map of GSE94752 with co-DEGs related to AF and obesity. Pink, obese samples; blue, lean samples; red, greater expression; green, less expression.

**Figure 3 biology-12-00121-f003:**
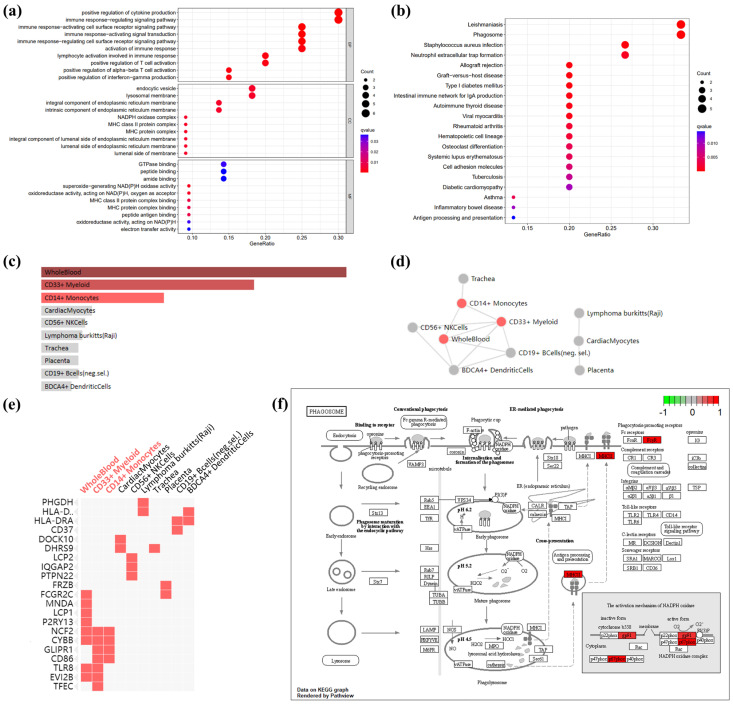
GO terms, KEGG pathway, and cell-type enrichment. (**a**) GO term enrichment for AF- and obese-related co-DEGs. Dot sizes, counts of enriched DEGs; dot colors, q value. (**b**) KEGG pathway of AF- and obese-related co DEGs. Dot sizes, counts of enriched DEGs; dot colors, q value. (**c**) Bar graph showing *p*-value for Enrichr cell-type enrichment, sorted by *p*-value ranking. Red nodes, adjust *p*-value < 0.05; gray nodes, adjust *p*-value > 0.05. (**d**) Network showing the interaction of enriched cell types. Red nodes, adjust *p*-value < 0.05; gray nodes, adjust *p*-value > 0.05. (**e**) Clustergram shows the relationship between cell-type enrichment and genes, sorted by *p* value ranking. Enriched terms are the columns, input genes are the rows, and cells in the matrix indicate if a gene is associated with a term. (**f**) KEGG pathway map of the phagosome, which is enrichment in co DEGs: *CYBB* (gp91), *NCF2* (p67phox), *HLA-DRA* (MHCII), and *FCGR2C* (FcγR). Red, up-regulated genes; green, down-regulated genes.

**Figure 4 biology-12-00121-f004:**
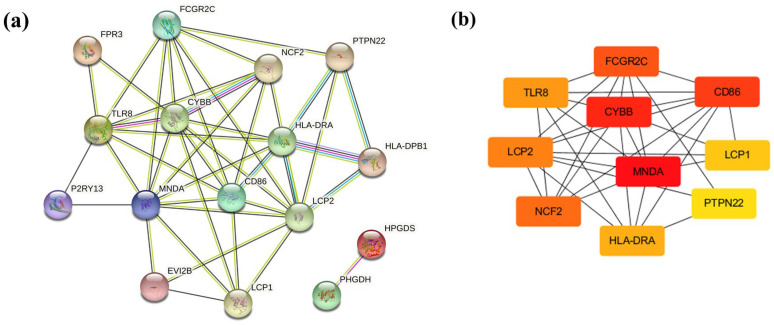
PPI network and hub genes. (**a**) PPI networks constructed by STRING database for co-DEGs (threshold > 0.4). (**b**) Hub genes were calculated by Cytoscape software. Red, higher MCC score; yellow, lower MCC score.

**Figure 5 biology-12-00121-f005:**
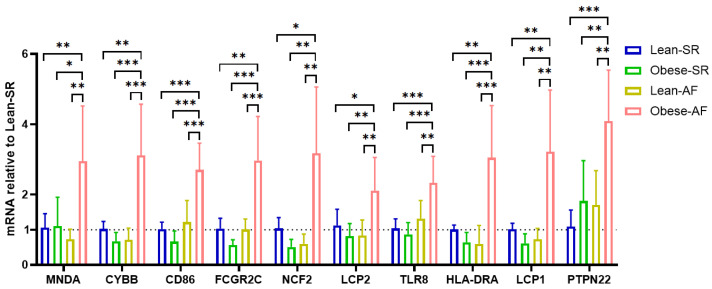
Validation of hub genes. Validation of hub genes in patients’ blood cells by qPCR, each of genes is relative to lean-SR group. * *p* < 0.05, ** *p* < 0.01, *** *p* < 0.001.

**Figure 6 biology-12-00121-f006:**
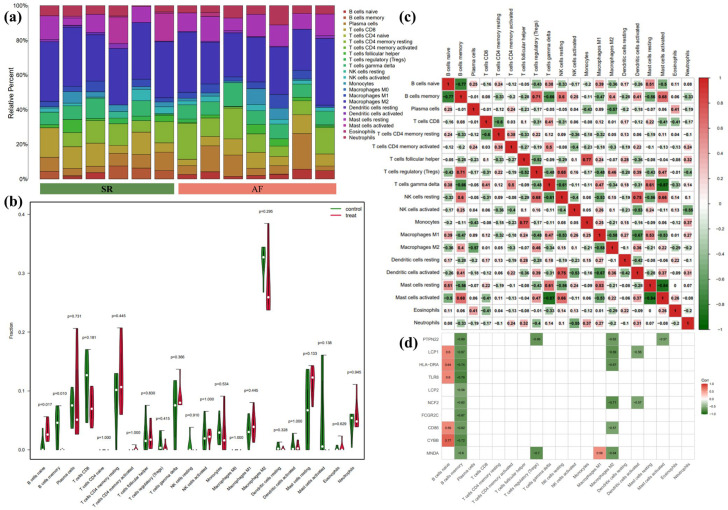
Results of CIBERSORT analysis of GSE79768 (AF-related array). (**a**) Cumulative histogram of immune cell infiltration. (**b**) Violin diagram of immune cell proportions in two groups. Green and fusiform fractions on the left, SR group; red and fusiform fractions on the right, AF group. (**c**) Correlation matrix of infiltration degree of immune cells. Red, positive correlation between two immune cells; blue, negative correlation between two immune cells. The bigger size of the numbers statistics data, the more positive or negative correlation. (**d**) Correlation matrix of infiltration degree of immune cells and hub genes. Red, positive correlation between two immune cells; blue, negative correlation between two immune cells.

## Data Availability

Data are available in a publicly accessible repository that does not issue DOIs. Publicly available datasets were analyzed in this study. This data can be found here: http://www.ncbi.nlm.nih.gov/geo/ (GSE79768 and GSE94752F, accessed on 26 April 2022).

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
