# Peer review of "Identification and Verification of Biomarkers and Immune Infiltration in Obesity-Related Atrial Fibrillation"

_biology, 2023, doi:10.3390/biology12010121_

Round 1
Reviewer 2 Report
The paper by Xie et al. titled “Identification and Verification of Biomarkers and Immune In-2 filtration in the Obesity-related Atrial Fibrillation” identified 10 genes were upregulated in obesity-related atrial fibrillation and these findings will help elucidate the pathophysiology and possible therapeutic targets of the disease. The manuscript is clear and the flow of the paper is good.
However, few corrections are required:
1. Please provide full form for SR.
2. Please include PMID: 25959929, PMID: 30918664
3. Please check the journal guidelines for the references.
Round 2
Reviewer 1 Report
The authors have answered all of my questions and the paper has been greatly improved. Therefore, it can be accepted for publication.
I have a final suggestion which does not compromised the acceptance of this paper. The authors should provide all figures in high-quality for publication.
Thank you